# Mixture-of-Queries Transformer: Camouflaged Instance Segmentation via Queries Cooperation and Frequency Enhancement

## Abstract

Due to the high similarity between camouflaged instances and the surroundings and the widespread camouflage-like scenarios, the recently proposed camouflaged instance segmentation (CIS) is a challenging and relevant task. Previous approaches achieve some progress on CIS, while many overlook camouflaged objects' color and contour nature and then decide on each candidate instinctively. In this paper, we contribute a Mixture-of-Queries Transformer (MoQT) in an end-to-end manner for CIS which is based on two key designs (a Frequency Enhancement Feature Extractor and a Mixture-of-Queries Decoder). First, the Frequency Enhancement Feature Extractor is responsible for capturing the camouflaged clues in the frequency domain. To expose camouflaged instances, the extractor enhances the effectiveness of contour, eliminates the interference color, and obtains suitable features simultaneously. Second, a Mixture-of-Queries Decoder utilizes multiple experts of queries (several queries comprise an expert) for spotting camouflaged characteristics with cooperation. These experts collaborate to generate outputs, refined hierarchically to a fine-grained level for more accurate instance masks. Coupling these two components enables MoQT to use multiple experts to integrate effective clues of camouflaged objects in both spatial and frequency domains. Extensive experimental results demonstrate our MoQT outperforms 18 state-of-the-art CIS approaches by 2.69% on COD10K and 1.93% on NC4K in average precision.

## 1 Introduction

Camouflage is a naturally evolved strategy for animals to hide themselves via adapting their body's coloring to match the surroundings, which is used for hunting prey or avoiding detection by natural enemies, as shown in Figure 1(a). Since there is a lot of demand for understanding the widespread camouflage-like scenarios, (*e.g.*, polyp segmentation Fan et al. (2020b), lung infection segmentation Fan et al. (2020c), search-and-rescue work Turić et al. (2010), manipulated image/video detection and segmentation Zhou et al. (2020)), the task of predicting the location and instance-level masks of camouflaged objects (*i.e.*, Camouflaged Instance Segmentation, CIS) has been proposed. Therefore, CIS is worth studying and has gradually received more attention in recent years. However, it also has challenges due to high intrinsic similarities between the target objects and the background.

Compared to the tremendous development in generic instance segmentation Bolya et al. (2019); Wang et al. (2020a;b); Ren et al. (2015); He et al. (2017a); Cai & Vasconcelos (2018); Chen et al. (2019), camouflaged instance segmentation remains an under-explored issue, and only a few efforts have been made to study it in the past three years Pei et al. (2022); Luo et al. (2023); Dong et al. (2023); Li et al. (2024); Le et al. (2023). CFL Le et al. (2021) is a first attempt. It is a two-stage method that fuses general instance segmentation methods for camouflaged instance segmentation but has limited performance. Subsequently, OSFormer Pei et al. (2022) is proposed as the first one-stage method for CIS. It takes advantage of a transformer network, which achieves a flexible framework that can be trained end-to-end for camouflaged instance segmentation. Recently, DCNet Luo et al. (2023) has been proposed to segment camouflaged instances via explicit de-camouflaging and achieves CIS by jointly modeling pixel-level camouflage decoupling and instance-level camou-

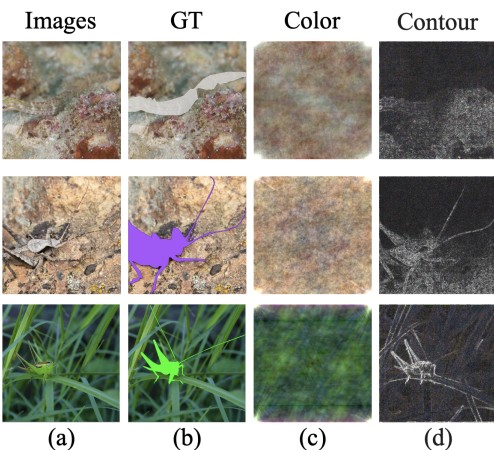

Images GT Color Contour

(a) (b) (c) (d)

Figure 1: The Nature of Camouflaged Objects. Careful contrast of the camouflaged inputs (a) and the corresponding ground truth (b), color (c) (reconstructed with only amplitude component of Fourier transformation) and contour (d) (reconstructed with only phrase component of Fourier transformation) information shows the important priori principle of camouflaged objects: Low-level statistics like color contain more information from the surroundings while high-level semantics like contour tend to preserve more camouflaged characteristics.

flage suppression. In the same period, UQFormer Dong et al. (2023) adopts a typical DETR-like architecture Carion et al. (2020) and exploits the information on object edges.

Although the aforementioned works have made some progress, two fundamental inspirations have not been taken into account: the priori of camouflage principles and the human habit of segmenting camouflaged instances. (**1**) The priori of camouflage principles: Only when you know how to camouflage can you see through camouflage. Many years ago, zoologists discovered that animals can camouflage themselves by matching their colors or patterns with the background. To look deeper into camouflage, we sample some images of camouflaged animals and analyze them thoroughly. Since it is hard to spot camouflaged objects in the surroundings, we perform the Fourier transform on these images to discover some clues in the frequency domain. We first decompose these camouflaged images into phrase and amplitude components and reconstruct images from only phrase component and amplitude component, respectively, presented in Figure 1. It is easy to find that the phase component of the Fourier spectrum preserves high-level semantics (contours and semantics) of original images, while the amplitude component contains low-level statistics (colors and styles). Therefore, enhancing the influence of contours and eliminating the interference of colors would certainly benefit the performance on CIS. (**2**) The human habit of segmenting camouflaged instances: When humans segment a camouflaged image, their visual system instinctively sweeps across the scene and determines some candidates. Then, the visual system gradually searches for valuable clues throughout the scene to obtain accurate segmentation masks. For some heavily camouflaged scenes with highly accurate segmentation like some medical image datasets Fu et al. (2019), it may even combine the masks labeled by multiple experts. Gradually refining and integrating the decisions of multiple experts are also potentially effective for CIS. Therefore, it makes sense to take full advantage of both inspirations of (**1**) and (**2**) for improving the performance of the CIS task.

Motivated by the above discussions, we proposed a Mixture-of-Queries Transformer (MoQT) trained in an end-to-end manner for CIS, which includes a Frequency Enhancement Feature Extractor (FEFE) based on modeling the colors and contours of camouflaged instances and a Mixture-of-Queries Decoder (MoQ Decoder) in transformer architecture referring to the segmentation process of multi-experts collaboration. First, inspired by the camouflage principles discussed above, we design a Frequency Enhancement Feature Extractor to capture more clues of camouflaged instances in the frequency domain. Specifically, we propose to adopt Fourier spectrum amplitude and phase to model image color information and contour information, respectively, as shown in Figure 1. With the help of color and contour information, we design a contour enhancement module and a color removal module, which can increase the contour effect while eliminating color interference. This mechanism in the frequency domain is suitable for debunking the principle of animal camouflaging, which is reasonable for providing gains on CIS. Second, for the Mixture-of-Queries Decoder, which is different from the standard DETR framework, we design multiple expert groups of queries according to the success of the Mixture-of-Experts (MoE). In transformer-based architecture, object queries are wonderful designs in transformer decoders, which have two roles: a) candidates for objects and b) interaction with transformer encoder features in the decoder layers for generating outputs. We propose a gating network on each decoder layer to mix multiple groups of queries

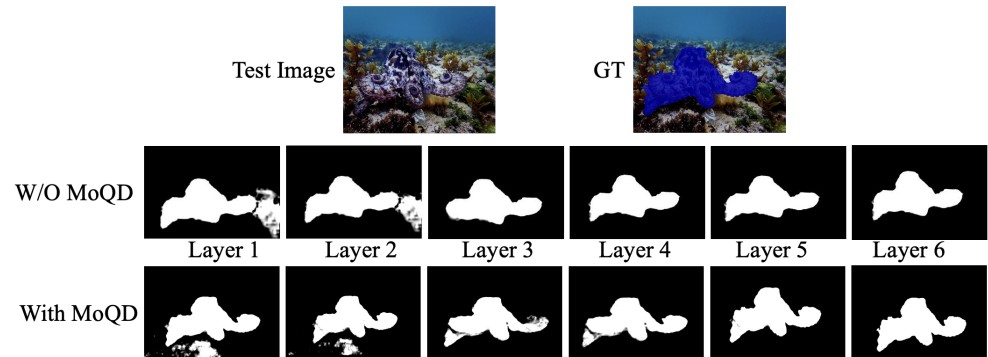

Figure 2: The Effectiveness of Mixture-of-Queries Decoder (MoQ Decoder). According to the comparison between the test image and ground truth, hierarchical prediction with MoQ Decoder can refine the output instance masks, including more accurate details than those without MoQ Decoder.

(several queries comprise an expert), deciding which experts are selected for the next layer forwarding. The gating network accepts the encoded features as input, and the final output of each layer is the weighted recombination of the various experts and the output of the gating network. This mechanism can refine outputs hierarchically to a fine-grained level via a mixture of experts, which can generate more accurate instance masks, as shown in Figure 2. In favor of these two designs, our method can utilize multiple expert queries to integrate effective clues of camouflaged objects in both spatial and frequency domains, which can achieve outstanding segmentation performance.

In summary, our main contributions are three-folds.

- We propose a Mixture-of-Queries Transformer (MoQT) trained in an end-to-end manner for CIS, which takes advantage of the priori of camouflage principles, and refers to the human habit of segmenting camouflaged instances.
- We proposed a Frequency Enhancement Feature Extractor (FEFE) and a Mixture-of-Queries Decoder (MoQ Decoder) for our MoQT, where FEFE is used for color removal and contour enhancement. The MoQ Decoder aims to mix multiple groups of queries hierarchically to provide more accurate predictions.
- Extensive experimental results on COD10K and NC4K show consistent performance gains compared with 18 baseline methods and verify the superiority of our method.

## 2 RELATED WORK

### 2.1 CAMOUFLAGED OBJECT DETECTION

Camouflaged Object Detection is usually considered as one of the most important origins of CIS and aims to identify the camouflaged objects from the background and has witnessed the development of art and biology Fan et al. (2020a); Le et al. (2019). Early research Huerta et al. (2007); Pan et al. (2011); Sengottuvelan et al. (2008) in COD mainly uses handcrafted features (*e.g.*, gradient, texture, and intensity features) to tell the camouflaged objects from their surroundings. Later, deep learning (DL) improves COD's performance in an end-to-end manner, and plenty of DL-based methods Pang et al. (2022); Yang et al. (2021); Piotrowski & Campbell (1982); Xu et al. (2021); Zhong et al. (2022); Mei et al. (2021); Ren et al. (2021) have been proposed. For example, ZoomNet Pang et al. (2022) discusses how to capture camouflaged objects in complex surroundings in a multi-scale manner. Moreover, UGTR Yang et al. (2021) combines the benefits of both Bayesian learning and transformer-based reasoning to handle camouflaged object detection with probabilistic and deterministic information. Some works Piotrowski & Campbell (1982); Xu et al. (2021); Zhong et al. (2022) even go beyond the RGB domain and explore frequency clues for better performance. In this paper, a Frequency Enhancement Feature Extractor, which refines frequency clues with contour enhancement and color removal, is adopted and allows full rein to both the camouflaged characteristics and the surrounding textures.

## 2.2 Camouflaged Instance Segmentation

Camouflage Instance Segmentation (CIS) learns most lessons from traditional instance segmentation. The purpose of instance segmentation is to assign pixel-level mask prediction for various instances. Nowadays, instance segmentation methods can be roughly divided into two parts: One-stage approaches Bolya et al. (2019); Wang et al. (2020a;b) and two-stage approaches Ren et al. (2015); He et al. (2017a); Cai & Vasconcelos (2018); Chen et al. (2019). Two-stage methods apply mask segmentation after proposal region detection, such as Faster R-CNN Ren et al. (2015), Mask R-CNN He et al. (2017a), Cascade R-CNN Cai & Vasconcelos (2018), and HTC Chen et al. (2019). CFL Le et al. (2021), the first attempt in CIS, also applies two-stage instance segmentation methods. However, one-stage methods show faster inference than two-stage methods and achieve comparable performance. For example, YOLACT Bolya et al. (2019) adopts two parallel tasks to produce non-local prototype masks with adaptive coefficients. Furthermore, SOLO Wang et al. (2020a) and SOLO-v2 Wang et al. (2020b) predict the instances' center and then decouple the instance masks with kernel feature learning. Recently, researchers have found transformers Cheng et al. (2021a; 2022b) show excellent performance on instance segmentation with the assistance of attention mechanisms and instance-specific prototypes. Therefore, transformer-based methods like OSFormer Pei et al. (2022), DCNet Luo et al. (2023) and UQFormer Dong et al. (2023) utilize transformers in CIS and achieve great progress. Inspired by Pei et al. (2022); Luo et al. (2023); Dong et al. (2023), our Mixture-of-Queries Transformer (MoQT) introduces a Mixture-of-Queries Decoder (MoQ Decoder) in the transformer decoder to combine the capabilities of multi-experts hierarchically, which enhances camouflage semantics and refines details of instance masks.

## 3 Method

### 3.1 Architecture Overview

The overall framework of our proposed model is presented in Figure 3. The whole architecture of our method is a typical MaskFormer-like Cheng et al. (2021b) model, composed of a Frequency Enhancement Feature Extractor (FEFE), a Pixel Decoder, and a Mixture-of-Queries decoder (MoQ Decoder). In the FEFE, we get valuable multi-scale features enhanced by the Fourier transform for revealing the camouflaged clues, where the phase component and amplitude can be used for modeling the information of contours and colors, respectively. We use a contour Enhancement Module (CEM) and a Color Remove Module (CRM) to mine the potential information of contours and eliminate the interference of colors for capturing clues of camouflaged instances. Then, the Pixel Decoder (based on FPN Lin et al. (2017)) gradually upsamples low-resolution features from the output of the backbone to generate high-resolution per-pixel embeddings. The MoQ Decoder computes from per-pixel embeddings and some initialized experts (a series of queries) to get the output prediction. Specifically, in MoQ Decoder, we propose a Mixture-of-Queries Layer (MoQ Layer) after each decoder layer and transform $M$ experts (each expert includes $N$ queries) via self and cross attention mechanisms, where the MoQ Layer is used to combine the $M$ experts of queries hierarchically. Finally, following previous work, we use a mask head and a matching algorithm to output the CIS prediction.

### 3.2 Frequency Enhancement Feature Extractor (FEFE)

As mentioned in our Introduction, the camouflage clues are mainly comprised of high-level semantics (*e.g.*, contours and semantics) and low-level statistics (*e.g.*, colors and styles), which can be reflected by the phrase and amplitude components of Fourier spectrum, respectively. As shown in Figure 1, it is believed that enhancing the influence of contours and eliminating the interference of colors would certainly benefit the performance of segmenting the camouflaged instances. Thus, to explore the camouflaged clues, we design FEFE to model the colors and contours of camouflaged objects, where the phrase and amplitude components are applied to identify the camouflaged semantics from surroundings in FEFE. Specifically, suppose $H$ and $W$ are the height and width of the input, and the Fourier transformation $\mathcal{F}(x)$ performed on each channel with a given camouflaged

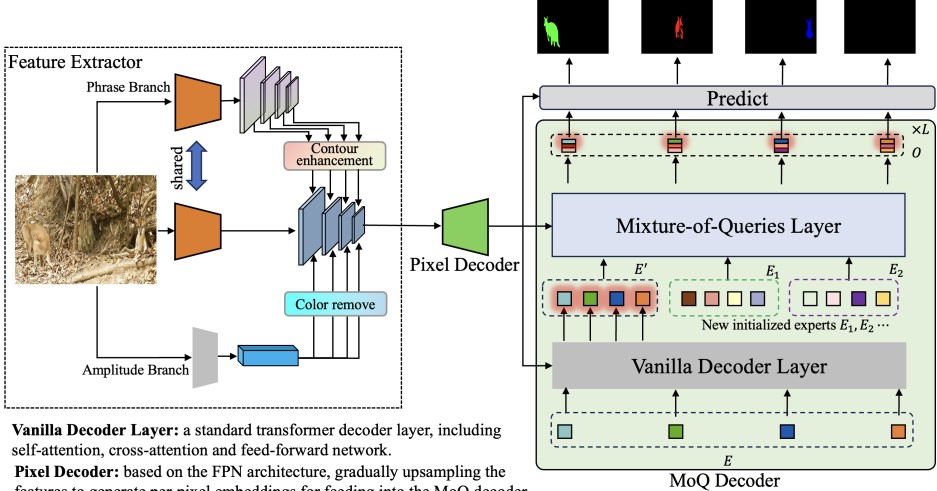

Figure 3: The Architecture of Our Proposed Model. Our method mainly consists of a Frequency Enhancement Feature Extractor (FEFE), a Pixel Decoder, and a Mixture-of-Queries Decoder (MoQ Decoder). (1) The FEFE captures suitable camouflaged clues with the contour enhancement and color remove modules in the frequency domain. (2) The Pixel Decoder is the same as previous works, based on the FPN architecture, which is used to gradually upsample low-resolution features from the output of the FEFE to generate high-resolution per-pixel embeddings. The detailed explanation of Pixel Decoder is presented in the *Appendix*. (3) The MoQ Decoder determines object candidates by multiple cooperation expert queries and hierarchically refines the instance masks with encoded features.

image $x \in \mathcal{R}^{3 \times H \times W}$ can be denoted as:

$$\mathcal{F}(x) = \sum_{i=0}^{H-1} \sum_{j=0}^{W-1} x[i,j] e^{-J2\pi(\frac{i}{H}u + \frac{j}{W}v)} = \mathcal{A}(x) e^{J\mathcal{P}(x)}, \tag{1}$$

where $J$ represents the imaginary unit, $\mathcal{A}(x)$ (modeling colors) and $\mathcal{P}(x)$ (modeling contours) are the amplitude and phrase components.

Then, we can get multi-scale image features $\mathbf{F}^k \in \mathcal{R}^{H \times W \times C}, k \in \{2,3,4,5\}$ extracted from a backbone network with the origin image $x$. Besides, we feed $\mathcal{A}(x)$ into a lightweight CNN and a $1 \times 1$ convolution to obtain the global camouflaged color information $\mathbf{F}_{color} \in \mathcal{R}^C$, and eliminate its interference via Color Remove Module (CRM). While for the phrase component $\mathcal{P}(x)$, due to that $\mathcal{P}(x)$ includes some information on contours and textures, extracting multi-scale features of $\mathcal{P}(x)$ can present unique advantages to mining camouflaged clues. Thus, we feed $\mathcal{P}(x)$ into the backbone and obtain hierarchical features $\mathbf{F}_{contour}^k, k \in \{2,3,4,5\}$ to explore the effects of contours as much as possible by the Contour Enhancement Module (CEM). Formally, the process of FEFE, including CRM and CEM for each scale feature $\mathbf{F} \in \{\mathbf{F}^2, \mathbf{F}^3, \mathbf{F}^4, \mathbf{F}^5\}$, can be expressed as:

$$\mathbf{F}_{refine} = \lambda \mathbf{F} \odot \mathbf{M}_{color} + (1 - \lambda) \mathbf{F} \odot \mathbf{M}_{contour},$$
$$\mathbf{M}_{color} = \delta \operatorname{Conv}(\operatorname{avg\_c}((\mathbf{F} - \mathbf{F}_{color})^2)), \tag{2}$$
$$\mathbf{M}_{contour} = \delta \operatorname{Conv}\left(\delta(\operatorname{MLP}(\operatorname{avg\_s}(\mathbf{F}_{contour}))) \odot \mathbf{F}\right),$$

where $\operatorname{avg\_c}$ and $\operatorname{avg\_s}$ indicate average pooling along spatial and channel axis, and $\delta$ is an activation function. CRM and CEM are designd to generate $\mathbf{M}_{color}$ and $\mathbf{M}_{contour}$, respectively. With the above module, we can get multi-scale refined features $\mathbf{F}_{refine}^k, k \in \{2,3,4,5\}$. Further, to acquire more fine-grained features for more accurate segmentation, we fuse $\mathbf{F}_{refine}^k, k \in \{2,3,4,5\}$ by feeding these features into the pixel decoder based on FPN Lin et al. (2017) architecture, which is used to gradually upsample low-resolution features from the output of the FEFE to generate high-resolution per-pixel embeddings $\mathcal{X} \in \mathcal{R}^{\frac{H}{4} \times \frac{W}{4} \times C}$.

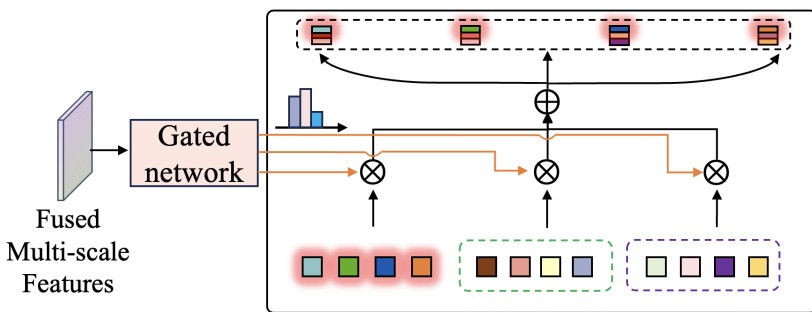

Figure 4: Illustration of the proposed Mixture-of-Queries Layer with a gated network to output the weights of each expert. The output is the weighted mixture of each input experts.

### 3.3 MIXTURE-OF-QUERIES DECODER (MoQ DECODER)

In order to capture camouflaged instances, the popular transformer-based architecture like Mask-Former Cheng et al. (2021b) and Mask2Former Cheng et al. (2022a), proposes a set of queries to identify whether each pixel belongs to a camouflaged instance. Meanwhile, as discussed in our Introduction, humans may segment camouflaged instances by gradually searching and multi-person collaboration. Inspired by the discussion, we propose a Mixture-of-Queries Decoder (MoQ Decoder) for hierarchically segmenting camouflaged instances.

#### 3.3.1 MIXTURE-OF-QUERIES MECHANISM

Different from the standard Transformer decoder architecture, in each layer, we introduce a Mixture-of-Quries Layer (MoQ Layer) after the original decoder layer, and initialize $M$ experts $E_i$, $i \in [1, M]$, where each expert contains $N$ queries $q_i$, $i \in [0, N-1]$, $q_i \in \mathcal{R}^d$. Each query is responsible for an object candidate. So, we have $E_i = \{q_0, q_1, \cdots, q_{N-1}\}$, $q_i \in \mathcal{R}^d$. Further, the detailed architecture of the designed MoQ Layer is illustrated in Figure 4, where the gated network $G$ outputs a sparse $(M+1)$-dimensional vector $G(x) \in \mathcal{R}^{M+1}$ to indicate the weights of each expert. Therefore, given the input of each decoder layer $E$ and the $M$ initialized experts $[E_1, E_2, \cdots, E_M]$, the output $y$ of the MoQ Layer can be written as follows:

$$y = \mathbf{E} \cdot \text{softmax}(G(\mathcal{X})), \tag{3}$$

where $\mathcal{X}$ is the output of pixel decoder, and $G(\mathcal{X})$ indicates the output of the gated network and $\mathbf{E} = [E', E_1, E_2, \cdots, E_M]$. $E'$ is the output of the standard transformer decoder layer fed with $E$. Besides, the forward process of each MoQ Decoder Layer (including a original decoder layer and a MoQ layer) can be formulated as:

$$
\begin{aligned}
Q &= W^Q \cdot E, \quad K = W^K \cdot \mathcal{X}, \quad V = W^V \cdot \mathcal{X}, \\
E' &= \text{LN}\left(E + \text{crossattention}(Q, K, V)\right), \\
E' &= [E', E_1, E_2, \cdots, E_M] \cdot \text{softmax}(G(\mathcal{X})), \\
O &= \text{LN}(E' + \text{MLP}\left(E'\right)),
\end{aligned}
\tag{4}
$$

where LN is layer normalization and MLP denotes the multi-layer perception network. During the training process, to provide deep supervision, we follow Mask2Former to adopt auxiliary losses with additional mask prediction heads and Hungarian match loss after each MoQ Layer.

#### 3.3.2 DISCUSSIONS

To elaborate further on our proposed MoQ Decoder, we provide detailed discussions about the differences between the MoQ Decoder and the vanilla transformer decoder. **1.** Compared with the vanilla transformer decoder, our MoQ Decoder does not contain just one group of queries for capturing various instances but multiple groups of queries in each MoQ Layer, (named as "expert" for each group of queries, noted as $E$, $E_1$ and $E_2$ in Figure 3). Benefiting from the success of MoE, there is a MoQ Layer for combining multi-experts of queries hierarchically to get a more accurate

Table 1: Performance Comparison of Various Methods. We display performance of state-of-the-art methods and our MoQT. The best results are in **bold**.

| Methods | COD10K-Test | | | NC4K-Test | | | Params(M) |
|---------|-----|-----------|-----------|-----|-----------|-----------|-----------|
| | AP | $AP_{50}$ | $AP_{75}$ | AP | $AP_{50}$ | $AP_{75}$ | |
| Mask R-CNN He et al. (2017b) | 25.0 | 55.5 | 20.4 | 27.7 | 58.6 | 22.7 | 43.9 |
| MS R-CNN Huang et al. (2019) | 30.1 | 57.2 | 28.7 | 31.0 | 58.7 | 29.4 | 60.0 |
| Cascade R-CNN Cai & Vasconcelos (2019) | 25.3 | 56.1 | 21.3 | 29.5 | 60.8 | 24.8 | 71.7 |
| HTC Chen et al. (2019) | 28.1 | 56.3 | 25.1 | 29.8 | 59.0 | 26.6 | 76.9 |
| BlendMask Chen et al. (2020) | 28.2 | 56.4 | 25.2 | 27.7 | 56.7 | 24.2 | 35.8 |
| Mask Transfiner Ke et al. (2022) | 28.7 | 56.3 | 26.4 | 29.4 | 56.7 | 27.2 | 44.3 |
| YOLACT Bolya et al. (2019) | 24.3 | 53.3 | 19.7 | 32.1 | 65.3 | 27.9 | - |
| CondInst Tian et al. (2020) | 30.6 | 63.6 | 26.1 | 33.4 | 67.4 | 29.4 | 34.1 |
| QueryInst Fang et al. (2021) | 28.5 | 60.1 | 23.1 | 33.0 | 66.7 | 29.4 | - |
| SOTR Guo et al. (2021) | 27.9 | 58.7 | 24.1 | 29.3 | 61.0 | 25.6 | 63.1 |
| SOLOv2 Wang et al. (2020b) | 32.5 | 63.2 | 29.9 | 34.4 | 65.9 | 31.9 | 46.2 |
| MaskFormer Cheng et al. (2021a) | 38.2 | 65.1 | 37.9 | 44.6 | 71.9 | 45.8 | 45.0 |
| Mask2Former Cheng et al. (2022b) | 39.4 | 67.7 | 38.5 | 45.8 | 73.6 | 47.5 | 43.9 |
| OSFormer Pei et al. (2022) | 41.0 | 71.1 | 40.8 | 42.5 | 72.5 | 42.3 | 46.6 |
| DCNet Luo et al. (2023) | 45.3 | 70.7 | 47.5 | 52.8 | 77.1 | 56.5 | 53.4 |
| UQFormer Dong et al. (2023) | 45.2 | 71.6 | 46.6 | 47.2 | 74.2 | 49.2 | 37.5 |
| CamoFourier Le et al. (2023) | 43.52 | **74.84** | 42.65 | 44.95 | 75.67 | 44.28 | - |
| MSPNet Li et al. (2024) | 39.7 | 69.8 | 39.8 | 41.8 | 71.8 | 42.3 | 48.09 |
| Ours | **47.99** | 73.01 | **51.77** | **54.73** | **78.45** | **58.97** | 61.68 |

prediction. **2.** Inspired by the human habit of segmenting camouflaged instances by gradual searches and multi-person cooperation. Each layer of MoQ Decoder always has $M$ newly initialized experts, indicating that different depths include different multi-experts responsible for capturing instances. While the vanilla transformer decoder just initializes one set of queries (*i.e.* one expert) in the first layer. Due to these two differences between the architecture of the standard transformer decoder and our proposed MoQ Decoder, our method can obtain a more accurate segmentation mask for each instance.

### 3.4 OBJECTIVE FUNCTION

As shown in Figure 3, with the fused feature $\mathcal{X} \in \mathcal{R}^{\frac{H}{4} \times \frac{W}{4} \times C}$ output by Pixel Decoder and the instance candidates $\hat{E} \in \mathcal{R}^{N \times C}$ generated by the MoQ Decoder, we can finally obtain the segmentation map, which can be formulated as:

$$\text{Mask} = \mathcal{X} \times \hat{E}. \tag{5}$$

To train the whole network, following DETR Carion et al. (2020), we adopt a Hungarian matching algorithm to match a ground truth label with each predicted segment instance. If no suitable label exists, a special label ("no object") is assigned. Therefore, including the instances and mask supervision, the objective function contains three terms: Cross-entropy Loss $\mathcal{L}_{CE}$ for the instance score, Focal Loss $\mathcal{L}_{focal}$ and Dice Loss $\mathcal{L}_{dice}$ for the mask predictions after each MoQ Layer, written as:

$$\mathcal{L}_{total} = \sum_{l=0}^{L} \mathcal{L}_{CE} + \alpha \cdot \mathcal{L}_{focal} + \beta \cdot \mathcal{L}_{dice}, \tag{6}$$

where $L$ means the amount of decoder layers. By default, we set $\alpha = 20$ and $\beta = 1$.

## 4 EXPERIMENTS

### 4.1 EXPERIMENTAL SETUPS

Following the mainstream works of CIS Dong et al. (2023); Luo et al. (2023), we evaluate our method in two datasets: COD10K and NC4K. COD10K includes 3040 training images and 2026 testing images, while NC4K contains 4121 test images for evaluating the generalization of proposed models. To provide a fair comparison, we train models in the training set in COD10K, and meanwhile test models in both test sets of COD10K and NC4K, which is a standard setting proposed in

Table 2: Performance Comparison of Proposed Modules. We perform an ablation study on COD10K and NC4K to validate our proposed modules' effectiveness. "FEFE" and "MoQ" represent Frequency Enhancement Feature Extractor and Mixture-of-Queries Decoder, respectively.

| FEFE | MoQ | COD10K-Test | | | NC4K-Test | | |
|------|-----|------|-----------|-----------|------|-----------|-----------|
| | | AP | $AP_{50}$ | $AP_{75}$ | AP | $AP_{50}$ | $AP_{75}$ |
| ✔ | | 45.76 | 71.33 | 49.31 | 53.22 | 77.84 | 57.29 |
| | ✔ | 47.12 | 72.76 | 50.61 | 53.87 | 78.02 | 58.12 |
| ✔ | ✔ | 47.99 | 73.01 | 51.77 | 54.73 | 78.45 | 58.97 |

Table 3: Performance Comparison of Various Backbones. We evaluate multiple methods' performance with various backbones on COD10K and NC4K.

| Method | Backbone | COD10K | NC4K |
|--------|----------|--------|------|
| OSFormer | | 41.0 | 42.5 |
| DCNet | ResNet-50 | 45.3 | 52.8 |
| Ours | | 48.0 | 54.7 |
| OSFormer | | 42.0 | 44.4 |
| DCNet | ResNet-101 | 46.8 | 53.5 |
| Ours | | 48.6 | 55.4 |
| OSFormer | | 47.7 | 50.2 |
| DCNet | Swin-Tiny | 50.3 | 56.3 |
| Ours | | 51.4 | 58.1 |
| OSFormer | | 52.1 | 56.7 |
| DCNet | Swin-Small | 52.3 | 58.4 |
| Ours | | 53.2 | 59.2 |

previous works Luo et al. (2023); Pei et al. (2022). In order to comprehensively evaluate the models, We use $AP_{50}$, $AP_{75}$, and $AP$ scores as evaluation metrics to quantify the performance of our method and baselines Luo et al. (2023); Dong et al. (2023); Pei et al. (2022). Besides, the implementation details of our experiments are presented in our *Appendix*.

## 4.2 COMPARISON WITH STATE-OF-THE-ART METHODS

The CIS task is a relatively novel task that has been proposed in recent years, and only a few previous works are involved in this task, such as OSFormer Pei et al. (2022), DCNet Luo et al. (2023), and UQFormer Dong et al. (2023). Consequently, we also adopt several popular generic instance segmentation methods as baselines on the CIS task for a more comprehensive test. And for a fair comparison, the backbone of these methods is configured as ResNet-50. The performance comparison results are shown in Table 1. It is easy to observe that our proposed model can consistently outperform the state-of-the-art methods by a large margin on both COD10K and NC4K test sets.

**(1) Results on COD10K.** As shown in Table 1, we compare our proposed model with 5 CIS models (*i.e.*, OSFormer Pei et al. (2022), DCNet Luo et al. (2023), UQFormer Dong et al. (2023), Camo-Fourier Le et al. (2023), and MSPNet Li et al. (2024)) and 13 generic instance segmentation models. Our model can achieve 51.77% in $AP_{75}$, which outperforms the second best method DCNet Luo et al. (2023) by 4.27% in $AP_{75}$. In $AP$, our model also gets a performance improvement of 2.69%. Notice that our method does not achieve the highest value in $AP_{50}$, instead of a comparable performance of 73.01% in $AP_{50}$. These results indicate that our method can acquire more accurate segmentation masks of camouflaged objects.

**(2) Results on NC4K.** Likewise, we evaluate these methods on NC4K dataset, and the results on this test set reflect the generalization ability of these models. Our model yields 58.97% in $AP_{75}$, while the previous best method DCNet is 56.5%, which demonstrates that our method gets an obvious gain of 2.47% in $AP_{75}$, suggesting a great generalization ability of our model as well. In AP, our model achieves the highest performance metrics of 54.73%, surpassing the second best method (DCNet) by 1.93%. Besides, our model also obtains a 1.35% improvement in $AP_{50}$. The overall metrics of various AP values reflect our method's obvious superiority over other baselines.

## 4.3 ABLATION STUDIES AND VISUALIZATIONS

To look deeper into our proposed method, in this section, we present a series of ablation studies to demonstrate the effectiveness of each proposed module.

**Effectiveness of proposed modules.** To explore the effectiveness of the proposed FEFE and MoQ Decoder, we validate the importance of each component by removing them one at a time. As shown in Table 2, the performance without MoQ Decoder drops by 2.23 % in $AP$, 1.68% in $AP_{AP_{50}}$ and 2.46% in $AP_{AP_{75}}$ on COD10K-Test. On NC4K-Test, the metrics of $AP$, $AP_{50}$ and $AP_{75}$ are also reduced by 1.51%, 0.66% and 1.68%, respectively. Similarly, if the components of FEFE are ablated, there is a drop in segmentation performance as well. For example, on COD10K-Test, the performance just achieves 47.12% in $AP$, 72.76% in $AP_{50}$ and 50.61% in $AP_{75}$, which are consistently lower than that without any modules ablated (as shown in the last row of Table 2). The

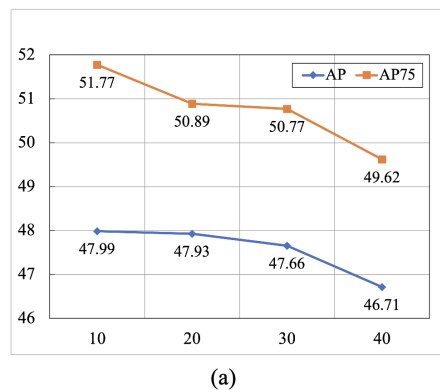 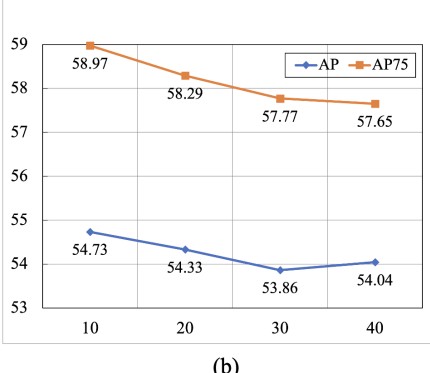

(a)                                           (b)

Figure 5: Performance Comparison of Various Numbers of Queries in Each Expert. AP and $AP_{75}$ of our MoQT with various numbers of queries on COD10K-Test (a) and NC4K-Test (b) are shown.

Table 4: Performance Comparison of Various Number of Decoder Layers. We apply various numbers of decoder layers, and the performance is shown as follows. The best results are in bold.

| Decoder Layers | COD10K-Test | | | NC4K-Test | | | Params(M) |
|---|---|---|---|---|---|---|---|
| | AP | $AP_{50}$ | $AP_{75}$ | AP | $AP_{50}$ | $AP_{75}$ | |
| 2 | 46.50 | 71.99 | 50.23 | 53.47 | 78.14 | 57.69 | 54.62 |
| 4 | 47.02 | 72.34 | 51.01 | 53.61 | 78.32 | 57.80 | 57.93 |
| 6 | **47.99** | **73.01** | **51.77** | **54.73** | **78.45** | **58.97** | 61.68 |
| 8 | 47.45 | 72.36 | 51.28 | 53.73 | 78.36 | 57.88 | 65.43 |
| 10 | 46.89 | 71.53 | 50.07 | 53.35 | 78.01 | 57.10 | 68.36 |
| 12 | 47.20 | 72.06 | 50.15 | 53.82 | 78.22 | 58.10 | 71.18 |

reduced performance demonstrates that these two proposed modules can capture clues of camouflaged instances and provide accurate segmentation. With both modules, our method can lead to huge performance gains in evaluation metrics.

**Various backbones.** To further explore the potential of our model, we equip it with different feature extractor backbones, such as ResNet-50 He et al. (2016), ResNet-101 He et al. (2016), SwinTransformer-Tiny (Swin-Tiny) Liu et al. (2021), and SwinTransformer-Small (Swin-Small) Liu et al. (2021). For a fair comparison with baselines, all these models are pretrained on ImageNet-1k Deng et al. (2009). The results are presented in Table 3. With the same backbone, our method shows the best performance among compared baselines, which indicates our method outperforms the state-of-the-art methods. For example, when ResNet-101 is the backbone, the metrics of $AP$ of our method are 48.6% and 55.4% on COD10K and NC4K datasets, respectively, while the second best method just reaches 46.8% and 53.5%. With a larger backbone, the results also prove that our method has the potential for further improvement.

**Ablation on the number of queries.** Object queries are essential in the transformer architecture for prediction. Therefore, we study the performance with different numbers of queries in each expert group. As shown in Figure 5, we change the number of queries from 10 to 40 and evaluate the performance metrics of $AP$ and $AP_{75}$ in both COD10K and NC4K test sets. In fact, the number of queries in each group should be larger than the actual count of objects to avoid instance fusion, which is determined by the dataset distribution. Moreover, it can be seen that when the number is set as 10, our model obtains the best performance on both datasets. For example, when the number is 10, $AP$ and $AP_{75}$ is 47.99% and 51.77% on COD10K, respectively. Meanwhile, $AP$ and $AP_{75}$ reach 54.73% and 58.97%.

**Analysis of the number of decoder layers.** We apply auxiliary losses after each decoding layer, as formulated in Equation (6). Hence, the number of decoder layers $L$ is important for the segmentation performance. As presented in Table 4, we vary the number of decoder layers, picked from the set $\{2, 4, 6, 8, 10, 12\}$. We find that the overall performance of the model improves with the increase of $L$. And when $L = 6$, the model can get the best performance. There is no additional performance

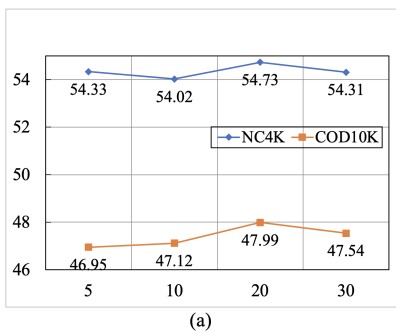
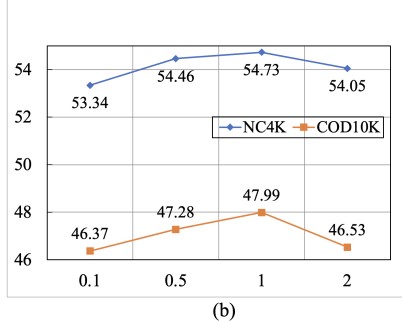

Figure 6: Ablation Studies on Hyper-parameters of $\alpha$ (a) and $\beta$ (b), presented in Equation (6).

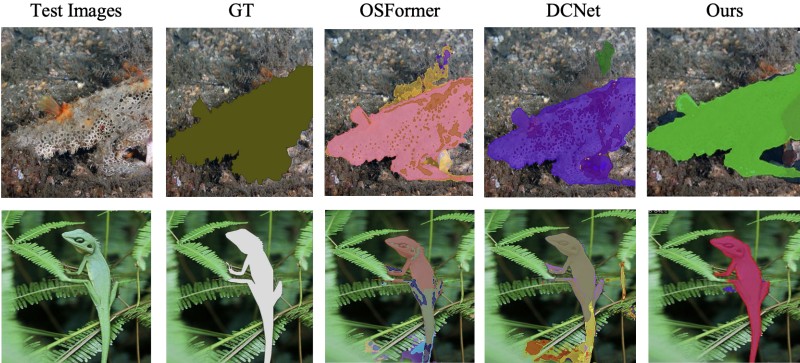

Figure 7: Visualizations of Various Methods. Different colored masks indicate different instances.

gain when the $L$ continues to increase, which may be caused by limited data to train the model for further improvement.

**Impacts about Hyper-parameters.** We study the impacts of the Hyper-parameters $\alpha$ and $\beta$ in Equation (6). On both COD10K and NC4K datasets, when $\alpha = 20$, the best performance is achieved, proved by the metrics of $AP = 47.99\%$ and $54.73\%$, respectively, shown in Figure 6(a). Therefore, we choose $\alpha = 20$ in our method by default. For the hyper-parameter $\beta$, we change the value of $\beta$ from 0.1 to 2, and the results as presented in Figure 6 (b). It can be seen that the model gets the best performance when $\beta = 1$. Therefore, to get the best performance, we set $\alpha$ as 20, and $\beta$ as 1.

**Visualization Results.** To comprehensively evaluate our method, we also present some qualitative analysis, as shown in Figure 7, referring to *appendix* for more visualization. We visualize the segmentation masks of various methods, including OSFormer Pei et al. (2022), DCNet Luo et al. (2023), and our method, to demonstrate the performance wtih qualitative results. It can be seen that our method performs better than previous methods, which can be proved by the clear boundaries and accurate masks of our method (shown in the last row of Figure 7). In short, our method not only improves the evaluation metrics on two datasets but also gains in visual results of segmentation masks.

## 5 CONCLUSION

In this paper, we propose a novel Mixture-of-Queries Transformer (MoQT) for camouflaged instance segmentation. MoQT applies a Frequency Enhancement Feature Extractor for feature extraction in the frequency domain, with the assistance of a contour enhancement module and a color removal module. Besides, a Mixture-of-Queries Decoder uses multiple expert groups of queries as candidates and shares semantic information with transformer encoder features. Multi-scale features enable MoQT to refine prediction hierarchically and get fine-grained instance masks with collaboration of multiple groups of queries. Compared with plenty of state-of-the-art baselines, our proposed MoQT shows outstanding performance on two benchmark datasets, demonstrating the proposed method's effectiveness.

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
