# A APPENDIX

## A.1 IMPLEMENTATION DETAILS

All experiments are implemented using the Pytorch framework Paszke et al. (2019) and trained on 4 NVIDIA GTX 3090Ti GPUs. Without additional explanation, the backbone of the feature extractor is ResNet-50 He et al. (2016) pretrained on ImageNet Deng et al. (2009), which is followed by previous works Luo et al. (2023); Dong et al. (2023); Pei et al. (2022). The parameters of the training phase are as follows. The batch size is configured as 4, and an Adam optimizer Kingma & Ba (2015) is chosen for training with an initial learning rate of 0.0001 for 100,000 iterations. The expert number of Mixture-of-Queries is 2 in each decoder layer, the query number of each expert is configured as 10, and the number of decoder layers is set as 6 by default.

## A.2 THE DETAILS OF PIXEL DECODER

To acquire fine-grained features for more accurate segmentation, we use multi-scale features $\{\mathbf{F}^i\}, i \in \{2, 3, 4, 5\}$ from different stage of the backbone. We feed the feature maps $(\mathbf{F}^2, \mathbf{F}^3, \mathbf{F}^4, \mathbf{F}^5)$ into the Pixel Decoder for fused features respectively. Specifically, our Pixel Decoder is based on the classical FPN Lin et al. (2017) and its details are illustrated in Figure 8. Thus, we can gradually upsample the features in a top-down pathway from lowest-resolution features, meanwhile aggregate features with the same resolution by lateral connections, and generate the high-resolution pixel-level features at $1/4$ scale of input image, which is used for final mask prediction.

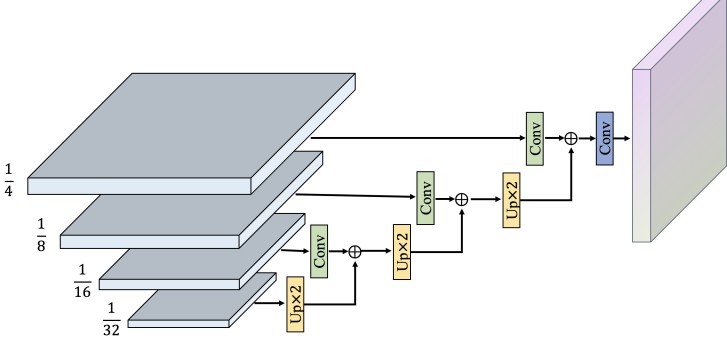

Figure 8: **Details of Pixel Decoder.** Our Pixel Decoder is based on the classical FPN Lin et al. (2017).

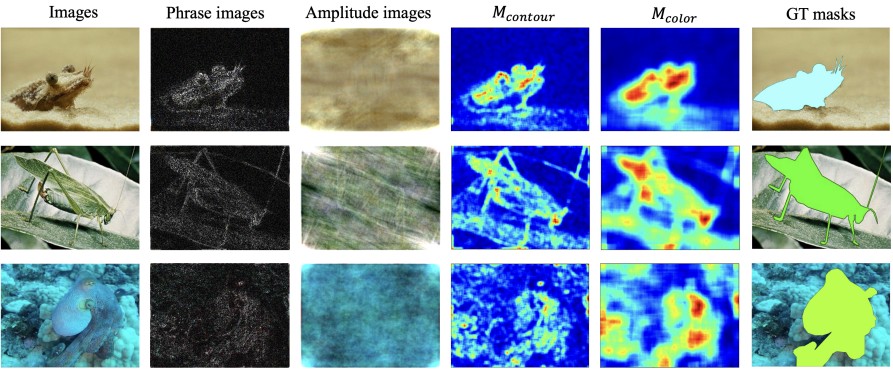

Figure 9: Visualization Results of CRM and CEM

### A.3 MORE VISUALIZATIONS

**Visualization of CRM and CEM.** To further explore the effectiveness of CRM and CEM in enhancing the influence of contours and eliminating the interference of colors, we present some visualization results for each module, as shown in Figure 9. With the help of both modules, our network can remove the confusing colors and localize the contours and textures (see 4th and 5th columns), which can facilitate final accurate segmentations.

**More qualitative results of various methods.** As presented in Figure 10, we provide more visualization results on predicted masks of various methods, including OSFormer Pei et al. (2022), DCNet Luo et al. (2023) and Ours. In terms of qualitative visualizations, our method performs the best among all methods.

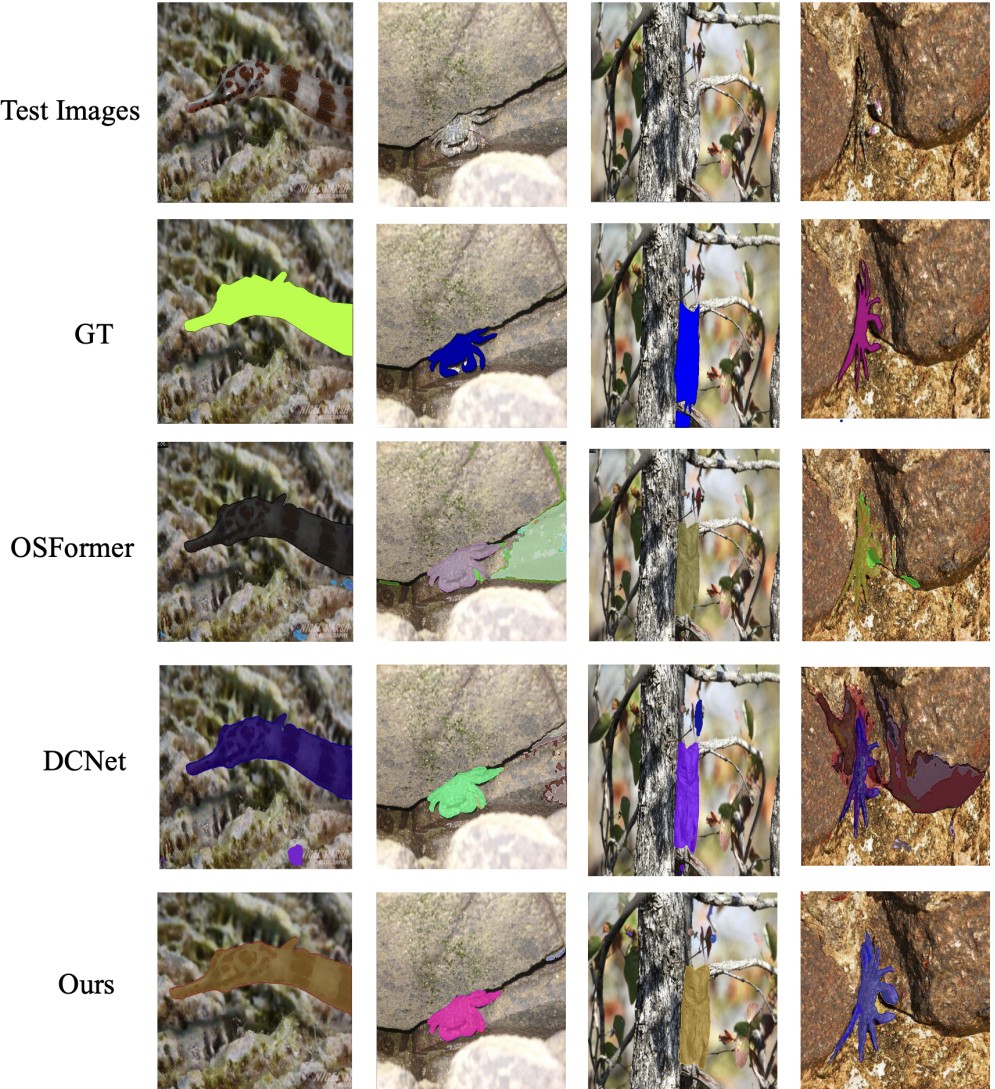

Figure 10: More Visualization Results on Mask Prediction of Various Methods.

**Visualization of failure cases and no camouflaged instances.** We present some extra visualizations about the failure cases, as shown in Figure 11 and no camouflaged instances 12. When the scene is very complex, our method fails to camouflaged instances. Because the camouflaged instances hide themselves heavily, our model and even humans can not distinguish them. And in the

Original image        Ground truth        Prediction

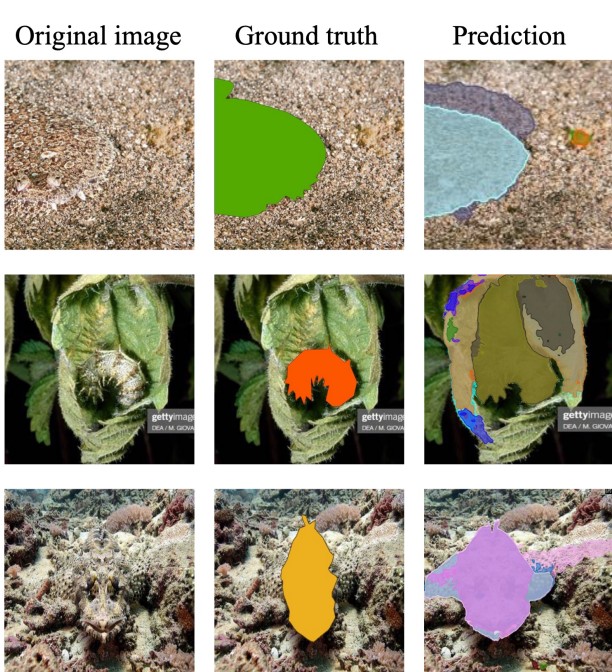

Figure 11: Visualization of failure cases.

No camouflaged image        Prediction        Ground truth

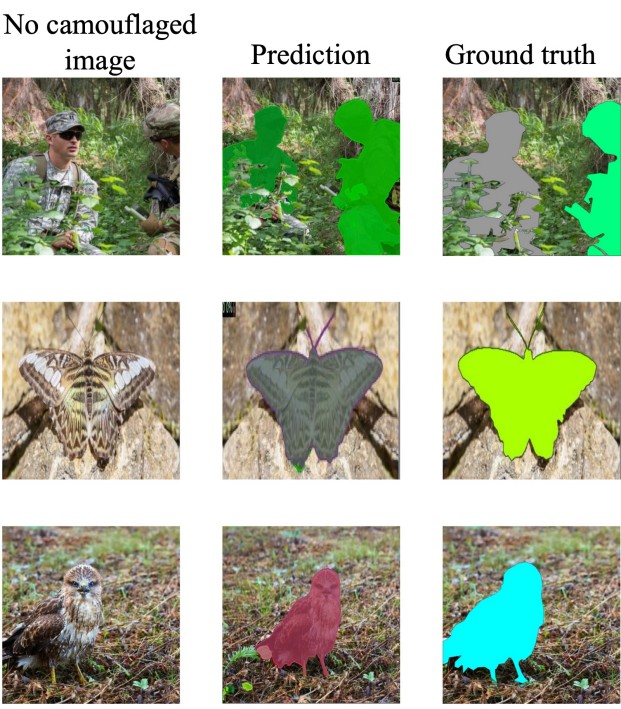

Figure 12: Visualization results on no camouflaged instances.

test set, there are some scenes can be recognized at a glance, not enough to be called camouflaged. And our model performs well on this no camouflaged instances.

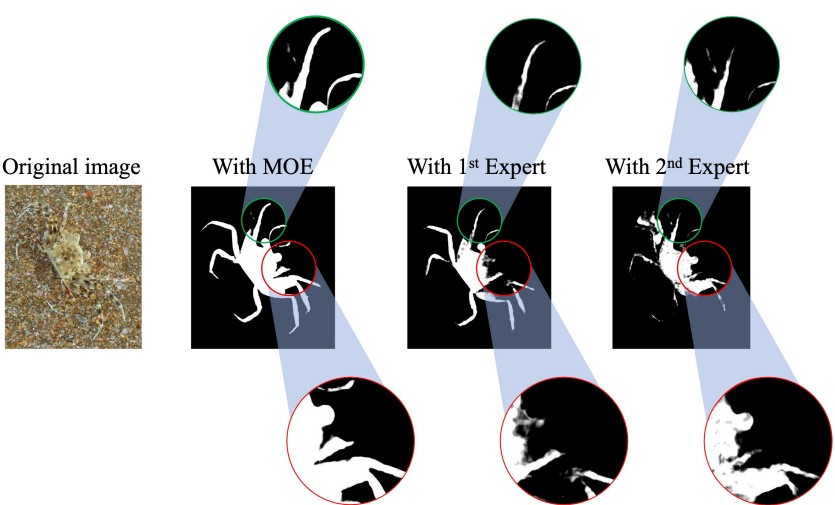

Figure 13: Visualization of various experts.

| Name | OSFormer | DCNet | UQFormer | CamoFourier | MoQT |
|---|---|---|---|---|---|
| Query-based strategy | DETR-like queries | DETR-like queries | Two sets of DETR-like queries | ❌ | Mixture-of-Queries (cooperation of multiple queries ) |
| Frequency components | ❌ | Amplitude | ❌ | Amplitude | Amplitude, phase |
| Insights | • First one-stage CIS work
• Corase-to-fine fusing | • Difference attention
• Reference attention | • Region and edge unified learning
• Joint learning | • Amplitude swapping
• Data augmentation | • Color removal and contour enhancement
• Revealing the relationship of Frequency and camouflage
• **First attempt** of MoE in query-based transformer |

Figure 14: Comparison of our MoQT and other CIS methods

**Visualization of various experts.**    we provide visualizations of various experts in Figure 13.  It can be found that with MoQ, the predicted masks are more accurate.  And various experts focus on various regions (the 1st expert focusing on green circle and the 2nd expert focusing on red circle) can be combined for accurate prediction masks.

A.4    MORE DISCUSSIONS ON CIS METHODS

We further discuss the difference between our method and other CIS methods, the comparison details are presented in Figure 14.  Difference from existing 4 methods (OSFormer, DCNet, UQFormer and CamoFourier), our proposed MoQT adopts color removal and contour enhancement in FEFE for mining camouflaged clues.  Besides, the MoQ decoder in our method is used to imitate the human habit of segmenting camouflaged instances, where in each layer we initialize new experts for cooperation and queries refining with MoE mechanism.  In summary, our method reveals the relationship of Frequency and camouflage, and it is the first attempt of using MoE mechanism in query-based transformer for segmentation.