# OpenReview forum: "Mixture-of-Queries Transformer: Camouflaged Instance Segmentation via Queries Cooperation and Frequency Enhancement"
_ICLR.cc/2025/Conference — ICLR 2025 Conference Withdrawn Submission_

### Official Review · Reviewer_fKVN · 2024-10-27

**Soundness:** 2
**Presentation:** 2
**Contribution:** 2
**Rating:** 3
**Confidence:** 5

**Summary:**

The paper presents a Mixture-of-Queries Transformer (MoQT) designed for camouflaged instance segmentation (CIS). It incorporates two main components: a Frequency Enhancement Feature Extractor (FEFE) and a Mixture-of-Queries Decoder. The FEFE captures camouflaged clues in the frequency domain by enhancing contours, eliminating interference colors, and extracting suitable features. The Mixture-of-Queries Decoder uses multiple experts of queries to spot camouflaged characteristics cooperatively, refining outputs hierarchically for accurate instance masks. Experimental results show that MoQT outperforms 18 state-of-the-art CIS approaches on COD10K and NC4K in average precision.

**Strengths:**

The approach shows significant improvement over existing methods, as demonstrated by extensive experimental results on COD10K and NC4K datasets, highlighting its practical applicability and potential impact in the field.

**Weaknesses:**

1. Lack of Innovation: The proposed frequency domain feature extraction method closely resembles those in "Unveiling Camouflage: A Learnable Fourier-based Augmentation for Camouflaged Object Detection and Instance Segmentation" and "Camouflaged Instance Segmentation via Explicit De-camouflaging." The Mixture-of-Queries Mechanism is also similar to the Multi-scale Unified Query Learning mentioned in "A Unified Query-based Paradigm for Camouflaged Instance Segmentation," without adequately explaining the main differences.

2. Writing and Expression Errors: There are several grammatical and expression errors in the manuscript. For example, lines 156-157 contain mistakes where "combines" should be "combine" and "camouflaged objection detection" should be "camouflaged object detection."

**Questions:**

1. On line 296 of the manuscript, when you mention initializing M experts E, are you referring to Positional Embeddings?

2. In line 320, it is stated that "our MoQ Decoder does not contain just one group of queries for capturing various instances but multiple groups of queries in each MoQ Layer." How are these groups designed and divided? The number of groups is not specified.

3. It is suggested to provide the code to help readers better understand the novelty of the proposed method.

---

> ### Author Response · Authors · 2024-11-27
> **response**
>
> **Anwser for Question1** : Thanks for your question. In line 296, our paper introduces the initialization of the experts  $E_i (i = 1, \cdots, M) $. These experts are groups of positional embeddings (i.e., queries in our manuscript) in the classical transformer. Referred from DETR, learned positional embeddings can also be named object queries. In summary, the experts are formed by the learnable positional embeddings (queries), which provide diverse information for camouflaged instance segmentation.
>
> **Answer for Question2**: Thanks for your question. We have specified the number of groups in Section A.1 (Implementation Details) of our Supplement Material. According to our supplement material, Mixture-of-Queries’s expert number (group number) is 2 in each decoder layer.
>
> **Answer for Question3**:  Thanks for your suggestion. The novelty of our method is illustrated in our paper and reply, and we propose to release our source code to the public after publication.
>
> **Answer for Weakness 1**: To clarify the novelty of our method, we further discuss the difference between our method and other CIS methods, the comparison details are presented in https://anonymous.4open.science/r/iclr2025_imgs-744F/table.jpg. Difference from existing 4 methods (OSFormer, DCNet, UQFormer and CamoFourier), our proposed MoQT adopts color removal and contour enhancement in FEFE for mining camouflaged clues. Besides, the MoQ decoder in our method is used to imitate the human habit of segmenting camouflaged instances, where in each layer we initialize new experts for cooperation and queries refining with MoE mechanism. In summary, our method reveals the relationship of Frequency and camouflage, and it is the first attempt of using MoE mechanism in query-based transformer for segmentation.
>
> **Answer for Weakness 2**: Thanks for your correction. We will carefully check our manuscript to avoid such mistakes in the final edition.

---

### Official Review · Reviewer_LLn2 · 2024-10-29

**Soundness:** 3
**Presentation:** 3
**Contribution:** 2
**Rating:** 5
**Confidence:** 3

**Summary:**

The paper proposes the Mixture-of-Queries Transformer (MoQT), a new model for camouflaged instance segmentation (CIS). The main contributions include: 1) Frequency Enhancement Feature Extractor (FEFE): This module leverages frequency-domain transformations to emphasize object contours and minimize color interference, aiding in detecting camouflaged instances by focusing on contour details rather than color; 2) Mixture-of-Queries Decoder (MoQ Decoder): This component employs multiple groups of object queries in a hierarchical framework, enhancing segmentation precision by refining masks at each layer. The model was benchmarked against 18 state-of-the-art CIS models and showed improved performance on the COD10K and NC4K datasets, with gains of 2.69% and 1.93% in average precision, respectively.

**Strengths:**

1. The paper introduces a novel approach to camouflaged instance segmentation (CIS) with the Frequency Enhancement Feature Extractor (FEFE) and Mixture-of-Queries Decoder (MoQ Decoder). These components creatively combine frequency-domain analysis and hierarchical query collaboration, offering a unique solution to the challenge of segmenting camouflaged objects.

2. The approach is rigorously validated, outperforming 18 state-of-the-art methods on key datasets. The paper's ablation studies and parameter analyses reinforce the model’s robustness and effectiveness, showcasing thorough and high-quality experimentation.

3. The paper is well-structured, clearly explaining its methods and their significance. Visual aids, including performance tables and diagrams, enhance understanding, and the rationale for each component is presented logically.

**Weaknesses:**

1. Insufficient Baseline Comparisons: While the paper includes comparisons with several CIS methods, it does not fully explore benchmarks with generic instance segmentation methods that could also apply to camouflaged segmentation. Including results for generic transformers or non-CIS-specific models with adaptations for camouflage (e.g., baseline Mask DINO [1] with FEFE added) would clarify the advantage of MoQT over generalized solutions.

2. The originality of Frequency Enhancement Feature Extractor (FEFE): The paper asserts that frequency domain-based contour enhancement is effective for CIS, but frequency domain analysis for camouflage object detection has already proposed by previous works like [2][3]. Although these works are dedicated for COD, the methodology of locate the camouflage objects is similar to CIS task.

3. Interpretability of the MoQ Decoder: Although the multi-expert query mechanism is innovative, the paper lacks insight into how each “expert” in the MoQ Decoder contributes uniquely to segmentation refinement. Visual or quantitative analysis of the individual contributions of each expert group in the MoQ Decoder would help to illustrate why this design is optimal and inform future work on multi-query designs.

[1] Mask DINO: Towards A Unified Transformer-based Framework for Object Detection and Segmentation, CVPR 2023

[2] Detecting Camouflaged Object in Frequency Domain, CVPR 2022

[3] Detecting Camouflaged Object in Frequency Domain, TOMM 2023

**Questions:**

1. Did the authors consider adapting standard segmentation models (e.g., Mask DINO, SAM) for CIS by incorporating frequency-domain enhancements like FEFE? If so, how did MoQT compare?

2. Why did the authors choose Fourier transforms over other frequency-based methods, such as wavelet transforms, for capturing contour information?

3. Can the authors visualize or quantitative analysis of the individual contributions of each expert group in the MoQ Decoder ?

4. Did the authors consider MoQT’s applicability to other segmentation tasks where objects are not necessarily “camouflaged” in the traditional sense?

---

> ### Author Response · Authors · 2024-11-27
> **response**
>
> **Answer for Question1 and Weakness1**: Thanks for your comments, and the answer is no.First we present **18 baselines in the paper**, including some typical segmentation methods, like Mask2Former and MaskFormer. Second, for fair comparison, **all the compared methods only use masks as supervision, and the backbone is pre-trained on ImageNet 1K, which are the same as previous works**. But your so-called standard segmentation models (e.g., Mask DINO, SAM) are not like this, even Mask DINO is not a standard segmentation model. **Mask DINO is a unified framework for detection and segmentation**, which adopts the same architecture design for detection as in DINO with minimal modifications (adopting a key idea from Mask2Former to construct a pixel embedding map). Further, Mask DINO is trained under the **supervision of boxes and masks, which is different from the 18 baselines**.  Mask DINO is **not fair** for comparison in CIS task. And we have present the comparison of Mask2Former in the paper, which is the one origin of Mask DINO. Besides, **SAM is pre-trained on large-scale datasets**, so it is also **unfair to show the comparison results with other baselines**. Therefore, we do not consider providing **extra unfair comparison results** with your so-called standard segmentation models.
>
> **Answer for Question2**: We prefer Fourier transform rather than Wavelet transform based on: **(1)** In general, **Fourier transform is more suitable for stationary signals and Wavelet transform for non-stationary signals**. With a given input, we consider that the distribution of the boundary of the camouflaged instances and the surroundings is stable with varying positions, which is adequate for stationary signals. **(2)** Fourier transform is **more convenient than Wavelet transform**.  Fourier transform can transform natural images into the frequency domain with only a few parameters and excellent performance. However, Wavelet transform needs much more hand-craft work (e.g., the selection of basic wavelet functions). **(3)** Fourier transform holds **a more tolerant attitude towards the structure of feature extractors**. On the one hand, Fourier transform performs consistently on frequency, and each frequency band’s contribution is equal in its output. The traditional feature extractors are convenient for processing such stationary signals. On the other hand, Wavelet transform has adaptive policies on high-frequency and low-frequency components, leading to unequal contributions of various frequency bands in its output.
>
> **Answer for Question3 and Weakness3**: As suggested, we provide visualizations of various  experts at https://anonymous.4open.science/r/iclr2025_imgs-744F/moe.jpg. It can be found that with MoQ, the predicted masks are more accurate, and various experts can combine them for better masks.
>
> **Answer for Question4**:  Thanks for your question.  (1) Our MoQT can also perform segmentation in traditional instance segmentation tasks. However, its mechanisms pay more attention to CIS, and its performance may not achieve state-of-the-art performance. Most previous works on CIS share the same characteristics since their improvement are task-specific. (2) Our MoQT is more likely to hold the SOTA performance in camouflaged-like segmentation tasks. The camouflaged-like segmentation means to split the target instance where the instance and surroundings are hard to distinguish. Since carefully designed mechanisms like FEFE and MoQ decoder are not only specific to the classical camouflaged scenarios, our MoQT probably performs well on these camouflaged-like instance segmentation tasks.
>
> **Answer for Weakness2**: Thanks for your question. We illustrate the originality of our FEFE in the following aspects: **(1) The purposes of  are different**. As mentioned in your question, these works focus on COD, and our MoQT pays attention to CIS.  **(2) The processes  are different**. On the one hand, our FEFE applies Fourier transform to the whole input and treats the amplitude and phrase components separately. Further, the contour enhancement on the phrase component and the color remove on the amplitude component are parallel. On the other hand, the previous works utilize discrete cosine transform (DCT) to the pre-processed 8 × 8 patches and split the frequency clues into several frequency bands. Besides, it performs band-wise and spatial-wise enhancement to these clues successively. **(3) The insights are different**. The previous works state that frequency clues are needed in COD and try to exploit the frequency clues for better performance in COD, while they reveal few explanations on why frequency clues can work. Our FEFE further points out that the phrase and amplitude components are responsible for high-level semantics (e.g., contour) and low-level semantics (e.g., color) in the camouflaged samples, providing valuable guidance to CIS. In summary, the originality of our FEFE is different from that of the previous works.

---

### Official Review · Reviewer_ApvJ · 2024-11-01

**Soundness:** 3
**Presentation:** 2
**Contribution:** 2
**Rating:** 6
**Confidence:** 4

**Summary:**

This paper aims to explore the influence of contour and color for discovering camouflaged instances and utilizes MoE technology to localize the multiple instances of camouflaged objects. First, the Frequency Enhancement Feature Extractor is proposed to capture the camouflaged clues in the frequency domain. To expose camouflaged instances, the extractor enhances the effectiveness of contour, eliminates the interference color, and obtains suitable features simultaneously. Second, a Mixture-of-Queries Decoder utilizes multiple experts of queries for spotting camouflaged characteristics with cooperation. The proposed MoQT achieves SOTA performance and outperforms 18 camouflaged instance segmentation methods on COD 10K and NC4K datasets.

**Strengths:**

1. This paper attempts a new respective to explore the concealed attribute of camouflaged instances: the visual cues of camouflaged objects are concealed, but the other domain (like frequency) cues of camouflaged objects are not completely hidden, which allows deep parsing of camouflage detection and segmentation. This idea is interesting and worth emulating.
2. The ablation experiments are sufficient.
3. The presentation of the paper is easy to understand, including some visual comparisons, etc.

**Weaknesses:**

1. The proposed MoQT employs the Fourier transform to obtain color and contour features. But introduction of Fourier transform is similar to the previous work CamoFourier:
Unveiling Camouflage: A Learnable Fourier-based Augmentation for Camouflaged Object Detection and Instance Segmentation, arXiV, 2023
2. Compared with the vanilla query-based decoder, the MoQ decoder introduces a Mixture-of-Queries (MoQ) layer with initialized M experts. Then, the M+1 outputs of the MoQ layer are aggregated via an adaptive weight. The novelty of MoQ is limited. The improvement of the query mechanism is a novel approach, but the proposed MoQ is only a token aggregation method. Besides, each layer of the MoQ decoder introduces initialized experts, which should hurt cross-attention enhanced query tokens. That sounds unreasonable.
3. The structure of Fig. 4 is not consistent with Eq. (4). The aggregated query tokens of multiple experts are input the vanilla query-based decoder in Eq. (4). But, the Fig. 4 does not present this process. Actually, the layer of MoQ decoder is twice the one of vanilla query-based decoder.

**Questions:**

1. I suggest that the author clearly compares this method with existing query-based transformer methods, and explicitly states the advantages and innovations of the method proposed in this paper.
OSformer: One-stage camouflaged instance segmentation with transformers. In European conference on computer vision, 2022
Camouflaged instance segmentation via explicit de-camouflaging. In Proceedings of the IEEE/CVF Conference on Computer Vision and Pattern Recognition, 2023
A unified query-based paradigm for camouflaged instance segmentation, ACM International Conference on Multimedia, 2023
2. I suggest that authors provide a more in-depth comparison with CamoFourier, highlighting any specific differences in how they utilize Fourier transforms and discussing how their approach advances beyond CamoFourier's techniques.
3. Authors should add some CIS methods from 2024 for comprehensive evulation, such as GLNet. Authors are suggested to directly compare their results with GLNet’s.
Camouflaged Instance Segmentation From Global Capture to Local Refinement, IEEE Signal Processing Letter, 2024.
4. The author should clearly articulate the number of query tokens used on each dataset and verify the impact of varied query token counts on different datasets.

In summary, the author should clearly elucidate the contributions of FEFE and MoQ to assess whether the paper meets the quality standards for acceptance at ICLR. For FEFE, simply using Fourier transform is not sufficient. For MoQ, aggregating multiple experis with query tokens and then inputing transformer decoder layers, this technology is not novel enough.

---

> ### Author Response · Authors · 2024-11-27
> **response**
>
> **Answer for Question1**: We further discuss the difference between our method and other CIS methods, the comparison details
> are presented in https://anonymous.4open.science/r/iclr2025_imgs-744F/table.jpg. Difference from existing 4 methods (OSFormer, DCNet, UQFormer and CamoFourier), our proposed MoQT adopts color removal and contour enhancement in FEFE for mining camouflaged clues. Besides, the MoQ decoder in our method is used to imitate the human habit of segmenting camouflaged instances, where in each layer we initialize new experts for cooperation and queries refining with MoE mechanism. In summary, our method reveals the relationship of Frequency and camouflage, and it is the first attempt of using MoE mechanism in query-based transformer for segmentation.
>
> **Answer for Question2 and Weakness1**: Thanks for your question. Although our MoQT and the CamoFourier apply Fourier transform in the CIS task, they hold different insights on the frequency clues. CamoFourier is based on the classical conditional GAN (c-GAN) structure and utilizes Fourier transform to synthesize transformed images for further object detection or instance segmentation. However, our MoQT uses Fourier transform to enhance contours and remove color information. Their differences are listed as follows: **(1) Different Purposes**: **CamoFourier uses Fourier transform for data augmentation** (also mentioned in its title "A Learnable Fourier-based Augmentation for ..."), but our **MoQT applies Fourier transform for color removal and contour enhancement**. **(2) Different Network Structures**: CamoFourier conducts a c-GAN structure with Fourier transform for data augmentation, but our MoQT simply utilizes a feature extractor (e.g., ResNet-50) with Fourier transform for feature enhancement. **(3)Different Insights**: CamoFourier aims to manipulate the amplitude information to enhance the visibility of camouflaged objects in the image, but our MoQT further considers both high-level semantics like contour (phrase information) tend to preserve more camouflaged characteristics and remove low-level statistics (amplitude information) like color contain more information from the surroundings.
>
> **Answer for Question3**: Thanks for you suggestion. For a fair comparison, we compare our method with GLNet as follows: with a swin'-tiny backbone, GLNet reaches 40.8@AP and 44.0@AP, and our method achieves 51.4@AP and 58.1@AP, which is much better than GLNet's performance. We will add the comparison results in the revised version. By the way, we find that in the original paper of GLNet, they compare the large backbone P2V (their method) with Resnet50 (baselines), which we think is unfair.
>
> **Answer for Question4**: Thanks for you comments. We have provide the results in Figure 5 in the submitted manuscript.
>
> **Answer for Weakness2**: The Mixture-of-Queries Decoder is proposed to imitate the human habit of segmenting camouflaged instances, and we initialize some experts for cooperation and queries refining hierarchically at each layer. In previous transformer-based methods, the decoder updates queries for final prediction. There is only one set of queries initialized before feeding into decoder layers, so they can only optimize the implicit relationship between the decoder and queries in an end-to-end manner. However, we initialize some experts at each layer to explicitly learn how to update queries. From the parameters learning aspect, our decoder can be regarded as a vanilla query-based decoder with some extra explicit parameters at each layer, which also play the same role as the parameters of original queries. Therefore, it does not hurt cross-attention but can provide more accurate refining. To our knowledge, the MoE mechanism of queries in transformer decoder has not been studied before, so we think this mechanism is novel.
>
> **Answer for Weakness3**: We are so sorry for your misunderstanding of our method due to the unclear presentation. In line 306,  ''MoQ Layer'' should be ''a layer of MoQ Decoder''. First, our MoQ decoder layer includes a vanilla query-based decoder layer and a MoQ Layer, and Eq(4) presents the process of a vanilla query-based decoder layer and a MoQ Layer.
> - The link of detailed illustration of MoQ Decoder: https://anonymous.4open.science/r/iclr2025_imgs-744F/moe_loc_eq.png

---

> > ### Comment · Reviewer_ApvJ · 2024-11-28
> >
> > The authors have explained the differences between the Fourier transform in CamoFourier and MoQT. They also clarified the novelty of the Mixture-of-Queries Decoder, which is proposed to imitate the human habit of segmenting camouflaged instances. Additionally, they have resolved the detailed issues raised. Therefore, I have raised my score.

---

### Official Review · Reviewer_rYM3 · 2024-11-04

**Soundness:** 3
**Presentation:** 3
**Contribution:** 2
**Rating:** 5
**Confidence:** 5

**Summary:**

In this paper, the authors tackle the problem of camouflaged instance segmentation (CIS). To this end, the authors proposed Mixture-of-Queries Transformer (MoQT). The experiments on COD10K and NC4K show that the MoQT outperforms other CIS baselines.

**Strengths:**

+ The paper is well-organized and easy to follow.
+ The proposed work outperforms 18 CIS baselines on COD10K and NC4K.
+ The authors did perform the ablation study to show the effectiveness of each component in MoQT.

**Weaknesses:**

+ There is a concern about the novelty. The authors explore frequency domain for feature extraction which is not new. The idea of using experts (several queries comprise an expert) is not new either.
+ The number of decoder layers, L, is questionable. There is a huge gap between 4 and 12. Why did the authors choose 6?
+ The visualization is not clear. How about the failure cases? How about the case of no camouflaged instance?

**Questions:**

+ I have question about the novelty.
+ There is a question about the parameters such as the number of decoder layers.

---

> ### Author Response · Authors · 2024-11-27
> **response**
>
> **Question1 (Weakness1)**: There is a concern about the novelty. The authors explore frequency domain for feature extraction which is not new. The idea of using experts (several queries comprise an expert) is not new either.
>
> **A1**: For the novelty, we clarify the following two aspects. **First**, we analyze in the introduction that our idea to solve CIS is based on **the characteristics of camouflage** (The priori of camouflage principles: zoologists discovered that animals can camouflage themselves by matching their colors or patterns with the background, but it is believed that there are some clues in contours for recognizing the camouflaged instances.) and **how humans distinguish camouflage objects** (The human habit of segmenting camouflaged instances: human visual system instinctively sweeps across the scene and gradually searches for valuable clues to find out camouflaged instances. And for some heavily camouflaged scenes it is believed that combining the masks labeled by multiple experts is much helpful.). The idea of solving CIS by **color removal and contour enhancement with the cooperation of multiple queries has not been proposed before**, and it makes sense according to the above analysis. **Second**, we propose a Mixture-of-Queries Transformer, which includes a Frequency Enhancement Feature Extractor (FEFE) and a Mixture-of-Queries Decoder (MoQ Decoder), where FEFE is used for color removal and contour enhancement. The MoQ Decoder aims to mix multiple groups of queries hierarchically to provide more accurate predictions. Although some previous methods introduce Fourier transform in camouflage scenes, **they do not reveal the relationship between the frequency domain and the camouflage scene**. Our proposed FEFE can **explicitly reveal the help and significance of frequency domain information in color removal and contour enhancement to expose the camouflaged object**, which is consistent with our analysis of the priori of camouflage principles in the introduction and not discussed before. Besides, the Mixture-of-Queries Decoder is proposed to imitate the human habit of segmenting camouflaged instances, and we initialize new experts for cooperation and query refining at each layer. In previous transformer-based methods, the decoder updates queries for final prediction. There is only one set of queries initialized before feeding into decoder layers, so they can only optimize the implicit relationship between the decoder and queries in an end-to-end manner. However, we initialize some experts at each layer to learn how to update queries. **Not only can each decoder layer optimize the candidates (refer to queries), but also the newly initialized experts can explicitly refine them hierarchically**, which is helpful for final predictions. To our knowledge, the MoE mechanism of queries in transformer decoder has yet to be studied. Moreover, we would appreciate it if you could find some similar works.
>
> **Question2 (Weakness2)** : There is a question about the parameters such as the number of decoder layers.
>
> **A2**: We have added extra detailed ablation results on decoder layers, as presented in the table (the metrics of AP are reported). It can be found that the best performance is achieved when the number of layers is 6. When the number of layers increases to 8, 10, and 12, the parameters become more, but the performance is not further improved, so we choose 6 as the decoder layers by default.
> | Decoder Layers | COD10K-Test | NC4K-Test | Params(M) |
> | ---- | ---- | ---- | ---- |
> | 2        | 46.50    | 53.47   | 54.62   |
> | 4        | 47.02    | 53.61   | 57.93   |
> | 6        | **47.99**  | **54.73** | 61.68 |
> | 8        | 47.45    | 53.73   | 65.43   |
> | 10       | 46.89    | 53.35   | 68.36   |
> | 12       | 47.20    | 53.82   | 71.18   |
>
> **Weakness3**: The visualization is not clear. How about the failure cases? How about the case of no camouflaged instance?
> **A3**: We present some extra visualizations of the failure cases and scenes with no camouflaged instance. Our method fails to segment the camouflaged instances from the surroundings when the scene is very complex. Because the camouflaged instances hide themselves heavily, our model and even humans find it difficult to distinguish them. In the test set, some scenes, recognized at a glance, are not enough to be called camouflaged. Our model performs well on these scenes with no camouflaged instances.
> - anonymized  link of failure cases: https://anonymous.4open.science/r/iclr2025_imgs-744F/fail.jpg
>
> - anonymized  link of no camouflaged instance:  https://anonymous.4open.science/r/iclr2025_imgs-744F/no.jpg

---

> > ### Comment · Reviewer_rYM3 · 2024-12-01
> >
> > After carefully considering authors' rebuttal and other reviewers' comments, I remain concerned about the paper's novelty. As a result, I have updated my score downward.

---

> > > ### Author Response · Authors · 2024-12-03
> > > **response**
> > >
> > > We found that you lowered your score from 6 to 5. After reading our rebuttal, can you tell us which part of the response increased your doubts?
> > >
> > > We can only guess where your doubts come from, which makes it difficult for us to give you a satisfactory answer. Just relying on one sentence of innovation concerns.
> > >
> > > According to your statement, you lowered the score based on our rebuttal and other people's opinions? This makes us also confused. Does your score depend on other people's opinions? If so, it is difficult for us to make a targeted response to your opinions, and it is not conducive to our in-depth discussion of the problems of the paper.

---

> ### Author Response · Authors · 2024-12-02
> **response**
>
> According to your replies,   I think the other problems have been solved by our rebuttal, but you still have some concerns about the novelty.  I can not agree with your comment of 'the authors explore frequency domain for feature extraction which is not new. The idea of using experts (several queries comprise an expert) is not new either'. **Please give us some evidence, we would appreciate it if you could find some similar works**.
>
> Besides, We further discuss the difference between our method and other CIS methods, the comparison details are presented in https://anonymous.4open.science/r/iclr2025_imgs-744F/table.jpg. Difference from existing 4 methods (OSFormer, DCNet, UQFormer and CamoFourier), our proposed MoQT adopts color removal and contour enhancement in FEFE for mining camouflaged clues. Besides, the MoQ decoder in our method is used to imitate the human habit of segmenting camouflaged instances, where in each layer we initialize new experts for cooperation and queries refining with MoE mechanism. In summary, our method reveals the relationship of Frequency and camouflage, and it is the first attempt of using MoE mechanism in query-based transformer for segmentation.
>
> Please **let us know where the flaws of our paper are, instead of just saying that it is not novel enough**, which is too cheap and unconvincing.

---

### Comment · Area_Chair_Se4N · 2024-11-30

Dear Reviewers,

Thank you again for your efforts in reviewing this submission. It has been some time since the authors provided their feedback. We kindly encourage you to review their responses, verify whether they address your concerns, and submit your final ratings. If you have additional comments, please initiate a discussion promptly. Your timely input is essential for progressing the review process.

Best regards,

AC

---

> ### Author Response · Authors · 2024-12-03
> **Response to AC**
>
> Dear AC,
>
> The participants in the discussion are not very active. It is really impossible to give a satisfactory response to the reviewers, and even impossible to have a complete conversation. I am also helpless.
>
> Even if there is a response, it is also some vague words, such as worrying about novelty. We cannot know the specific problems of the paper. We can only guess the focus of the reviewers and give answers, which is not conducive to discussion.

---

### Note · Authors · 2025-01-16

I have read and agree with the venue's withdrawal policy on behalf of myself and my co-authors.